# The Role of Epigenetics in Brain Aneurysm and Subarachnoid Hemorrhage: A Comprehensive Review

**DOI:** 10.3390/ijms25063433

**Published:** 2024-03-19

**Authors:** Isabel Fernández-Pérez, Adrià Macias-Gómez, Antoni Suárez-Pérez, Marta Vallverdú-Prats, Eva Giralt-Steinhauer, Lidia Bojtos, Sílvia Susin-Calle, Ana Rodriguez-Campello, Daniel Guisado-Alonso, Joan Jimenez-Balado, Jordi Jiménez-Conde, Elisa Cuadrado-Godia

**Affiliations:** 1Neurovascular Research Group, Hospital del Mar Research Institute, 08003 Barcelona, Spain; ifernandezperez@psmar.cat (I.F.-P.); amaciasgomez@psmar.cat (A.M.-G.); asuarezperez@psmar.cat (A.S.-P.); mvallverdu@researchmar.cat (M.V.-P.); egiralt@imim.es (E.G.-S.); arodriguezc@psmar.cat (A.R.-C.); dguisadoalonso@psmar.cat (D.G.-A.); jjimenez3@researchmar.net (J.J.-B.); jjimenez@researchmar.net (J.J.-C.); 2Neurology Department, Hospital Del Mar, 08003 Barcelona, Spain; lbojtos@hmar.cat (L.B.); ssusincalle@hmar.cat (S.S.-C.); 3Faculty of Medicine and Life Sciences, Universitat Pompeu Fabra, 08003 Barcelona, Spain

**Keywords:** subarachnoid hemorrhage, intracranial aneurysm, epigenetics, DNA methylation, microRNA, circular RNA, long non-coding RNA

## Abstract

This comprehensive review explores the emerging field of epigenetics in intracranial aneurysm (IA) and aneurysmal subarachnoid hemorrhage (aSAH). Despite recent advancements, the high mortality of aSAH needs an understanding of its underlying pathophysiology, where epigenetics plays a crucial role. This review synthesizes the current knowledge, focusing on three primary epigenetic mechanisms: DNA methylation, non-coding RNA (ncRNA), and histone modification in IA and aSAH. While DNA methylation studies are relatively limited, they suggest a significant role in the pathogenesis and prognosis of IA and aSAH, highlighting differentially methylated positions in genes presumably involved in these pathologies. However, methodological limitations, including small sample sizes and a lack of diverse population studies, temper these results. The role of ncRNAs, particularly miRNAs, has been more extensively studied, but there are still few studies focused on histone modifications. Despite methodological challenges and inconsistent findings, these studies underscore the involvement of miRNAs in key pathophysiological processes, including vascular smooth muscle regulation and the inflammatory response. This review emphasizes methodological challenges in epigenetic research, advocating for large-scale epigenome-wide association studies integrating genetic and environmental factors, along with longitudinal studies. Such research could unravel the complex mechanisms behind IA and aSAH, guiding the development of targeted therapeutic approaches.

## 1. Introduction

Aneurysmal Subarachnoid hemorrhage (aSAH) is a cerebrovascular disease caused by the rupture of an intracranial aneurysm (IA). Although it constitutes only 5% of all strokes, it has a significant socioeconomic impact being an important cause of acquired brain damage in young individuals, especially in women [1,2]. Despite therapeutic advancements in recent years, aSAH remains a considerable challenge, such that the mortality rate is 30% and one-third of survivors experiencing severe sequelae, including physical disability and cognitive impairment [2]. It represents, therefore, a compelling health problem.

Key risk factors associated with the formation and rupture of IAs include female sex, smoking, and hypertension [3]. Furthermore, there is a significant genetic predisposition, with the risk of both IA and aSAH increasing according to the number of first-degree relatives affected in the family [4]. However, genetic variation seems to account for only 40% of the disease heritability, leaving a substantial portion of the risk unexplained [5]. As a result, environmental factors have been proposed as contributors to the heightened familial risk, potentially mediated through epigenetic modifications [3].

The prognosis of aSAH depends on two critical factors: the initial bleeding severity, assessed with clinical or radiological scales, and the occurrence of medical complications. Among these complications, rebleeding of the aneurysm and delayed cerebral ischemia (DCI) are the most severe ones. While advances in healthcare systems and endovascular therapies have markedly reduced the incidence of aneurysmal rebleeding, DCI remains a significant challenge, afflicting 30% of cases [6]. The primary risk factor for DCI is arterial vasospasm (VS), which typically manifests between the third and tenth day after the initial hemorrhage. Nevertheless, clinical studies and randomized clinical trials have revealed that VS does not fully explain the risk of DCI, which might be triggered by alternative pathophysiological pathways, such as microthrombosis, inflammation, and cortical spreading depression [7,8,9].

Recent years have witnessed an increased interest in studies exploring the genetic and epigenetic interactions underlying the pathogenesis of IA and aSAH. While the genetic advances have been summarized elsewhere [10], this comprehensive review seeks to elucidate the current state of knowledge regarding the role of epigenetics in the development of IA and aSAH, including the emergence of complications such as VS and DCI as well as the prognosis.

## 2. Overview of Epigenetics and Its Mechanisms

Epigenetics is the study of changes in gene expression or cellular phenotype that do not involve modifications to the underlying DNA sequence. Epigenetic mechanisms that regulate gene expression are usually heritable, but they are also influenced by the environment, lifestyle, and other external factors [11]. These mechanisms are fundamental to the regulation of many cellular and biological processes, and several diseases are known to be caused or affected by diverse epigenetic modifications. To date, several epigenetic processes have been described, although they are usually grouped into three main categories: DNA methylation, histone modifications, and non-coding RNA.

### 2.1. DNA Methylation (DNAm)

DNAm is the most broadly studied epigenetic mechanism, which is defined as the addition of a methyl group in a Cytosine–Phosphate–Guanine (CpG) dinucleotide by enzymes called DNA methyltransferases. While only 1% of the total human genome contains CpG sites, most of them are clustered in areas called CpG islands, which are present in 70% of genes, especially near promoter regions [11,12,13]. DNAm can have complex and context-dependent effects on gene expression. DNAm occurring in gene bodies, for example, can either increase gene transcription or result in transcription suppression. In addition, DNAm is involved in regulating the expression of a large number of genes, including genes that control the development of cells and tissues. Moreover, DNAm varies with ageing, and changes at specific CpG sites allow the construction of biological age estimators, known as epigenetic clocks. The difference between chronological and biological age, referred to as age acceleration, appears to be a good predictor for a broad range of health outcomes [14,15].

DNA methylation is also involved in pathological processes, regulating the expression of genes related to cancer and many other diseases [11]. In the recent decades, improvement in microarray technologies and the popularization of epigenome-wide association studies (EWAS) have appeared as a new paradigm that offers a global view of the methylome (DNAm), in a similar manner as genome-wide association studies (GWAS). This approach captures relevant differential DNAm signatures associated with a phenotype of interest. For instance, regarding cerebrovascular diseases, previous studies have demonstrated the role of DNAm in the pathogenesis of ischemic stroke, intracerebral hemorrhage, and subclinical cerebrovascular disease [16,17,18]. However, the contribution of DNAm in IA and aSAH has been less studied so far.

### 2.2. Non-Coding RNA Molecules (ncRNAs)

Recent genomic studies have shown that despite comprising over 98% of non-protein-coding sequences, the human genome is extensively transcribed, producing a wide range of ncRNA subtypes [19]. These ncRNAs represent a complex and dynamic regulatory network that extends far beyond the traditional view of protein-coding genes and play pivotal roles in the regulation of nearly every cellular process [20]. Among the different ncRNA subtypes, microRNAs (miRNAs), long non-coding RNAs (lncRNAs), and circular RNAs (circRNAs) have been recently characterized.

#### 2.2.1. MicroRNAs

miRNAs represent one of the most extensively studied subclasses of ncRNAs. They are endogenous short (typically consisting of 18–25 nucleotides) RNA molecules that regulate the gene expression of approximately 60% of human protein-coding genes at the post-transcriptional level, primarily through the interaction with messenger RNAs (mRNAs) [21]. miRNAs recognize specific binding sites located within the 3′ untranslated region (3′ UTR) of mRNAs, thereby modulating either translational repression or, less frequently, mRNA degradation [22]. They exhibit a widespread expression across various tissues, with notably abundant levels in the central nervous system. Additionally, miRNAs have been identified in biofluids, such as human serum, plasma, and cerebrospinal fluid (CSF), packaged within microvesicles. The detection of miRNAs in biofluids offers potential insights into acute pathophysiological processes, characterized by their dynamic expression changes [23].

Importantly, miRNAs play a pivotal role in the regulation of key pathophysiological processes that have been associated with IA and aSAH, including apoptosis, neuroinflammation, oxidative stress, angiogenesis, vascular integrity, and vascular function [24,25,26,27]. And therefore, in recent years, a growing body of research has delved into the specific roles of miRNAs in the context of IA and aSAH.

#### 2.2.2. Long Non-Coding RNAs

lncRNAs are non-protein-coding transcripts larger than 200 nucleotides. lncRNAs can interact with DNA, mRNA, protein, and miRNA, regulating gene expression at the transcriptional, post-transcriptional, translational, and post-translational levels. As a result of these interactions, lncRNAs participate in wide varieties of processes such as chromatin remodeling, transcriptional activation, transcriptional interference, RNA processing, and mRNA translation [28].

In recent years, there has been an increasing interest in studying the role of lncRNAs in human disease, because of its potential clinical applications and advantages; they are expressed in peripheral blood and have tissue and spatiotemporal specificity, making them useful biomarkers of many medical conditions [29]. Notably, lncRNAs have emerged as key players in various vascular and neurological diseases, including the pathogenesis of aortic aneurysms and ischemic stroke [30,31]. Furthermore, their potential as therapeutic targets has gained considerable attention, with ongoing investigations exploring their utility in drug therapies, particularly in the context of neurological tumors [32].

#### 2.2.3. Circular RNAs

circRNAs are a class of non-coding RNAs characterized by their closed-loop, single-stranded structure without free 3′ or 5′ ends. This circular conformation makes circRNA molecules highly resistant to degradation by ribonuclease R, leading to a high stability and abundance compared to lncRNAs. circRNAs exert their regulatory influence on gene expression at both the transcriptional and post-transcriptional levels, engaging in various mechanisms [33].

One prominent role of circRNAs is their capacity to function as miRNA sponges, sequestering miRNAs away from their target mRNAs, thus relieving miRNA-mediated gene suppression. Additionally, circRNAs can act as platforms for RNA-binding proteins, influencing their activity and interactions. Furthermore, circRNAs participate in the modulation of alternative splicing and parental gene expression, contributing to the complex modulation of gene expression [33,34].

circRNAs represent a relatively recent discovery in the field of ncRNAs, and research into their functional relevance continues to evolve. Their involvement has already been linked to numerous vascular diseases, including atherosclerosis [35] and aortic aneurysms [36], underscoring their emerging significance in the field of molecular biology and disease pathology.

### 2.3. Histone Modification

Histones are proteins that package and compact DNA into a structure called chromatin. Histone modifications can cause the chromatin to become more compact or relaxed, influencing the accessibility of the DNA to the cellular machinery responsible for gene expression. Currently, more than ten types of post-translational modifications have been described, the main four including acetylation, methylation, phosphorylation, and ubiquitination, occurring on basic histone residues such as arginines, lysines, and histidines. The functional implications of these modifications are highly dependent on the specific residue that undergoes alteration. For example, methylation at lysine 4 and 27 on histone 3 (H3K4 and H3K27) is associated with transcriptional activation or silencing, respectively. Techniques such as Chromatin Immunoprecipitation followed by sequencing (ChIP-Seq) are employed for localizing the histone modifications in the genome. ChIP-Seq combines immunoprecipitation with advanced DNA sequencing, providing a comprehensive genomic distribution map of specific histone modifications [12].

These emerging insights into epigenetic regulation highlight its significant impact on human health, promising advances in diagnostics, therapeutics, and disease prevention. Epigenetic research, focusing on the genetic–environmental interplay in complex disease pathogenesis, is particularly vital. Moreover, developing therapies targeting epigenetic modifications, like histone deacetylase inhibitors [34] and DNA methyltransferase inhibitors [28], represent a rapidly evolving research area with the potential for diverse medical applications.

## 3. Results

### 3.1. DNA-Methylation

We identified a modest number of studies focusing on DNAm and its association with the risk of IA formation, rupture, and complications of aSAH. These studies are often constrained by small sample sizes and lack of replication across independent and multiethnic populations, limiting the generalizability of results.

Regarding the IA formation risk, some studies have examined candidate-gene approaches, analyzing blood DNAm in genes previously implicated in IA pathophysiology. Notably, these studies found differential methylation levels in certain genes’ promoter regions, such nitric oxide synthase 1 adaptor protein (*NOS1AP*), platelet-derived growth factor-D (*PDGFD*), and mitogen-activated protein kinase kinase kinase 10 (*MAP3K10*), when comparing IA cases to healthy controls [37,38,39]. However, these findings lacked replication and functional analysis. Another study compared DNAm and gene expression between nine IA tissues and matched controls, identifying over 11,000 differentially methylated sites associated with pathways related to inflammation and smooth muscle cell activity [40]. Still, these findings lacked replication and correction for multiple testing, casting uncertainty on their reliability. Additionally, a fourth study found lower DNAm levels and a higher mRNA expression of the glutathione S-transferase alpha 4 *(GSTA4*) gene involved in oxidative stress and phenotypic vascular smooth muscle cells (VSMC) transition, which is associated with IA risk, particularly in women [41].

In the context of complications from aSAH, one study found a specific CpG site, hypermethylation associated with a decreased mRNA expression of insulin receptor (*INSR*) and cadherin-related family member 5 (*CDHR5*) genes in DCI cases, which was subsequently replicated in a larger cohort [42]. Another study did not identify significant CpGs associated with DCI but found suggestive associations with Angiopoietin 1 (*ANGPT1*), a gene with critical functions in angiogenesis after vascular injury [43].

Concerning aSAH prognosis, one study explored methylation signatures of Apolipoprotein E (*APOE*) and Factor XIIIA (*F13A*) genes in relation to both aSAH risk and outcome [44]. *F13A* is an important enzyme in the blood coagulation system and *APOE* is involved in regulating inflammatory responses, and both have been associated with aSAH risk [45,46]. They discovered hypermethylation of the F13A promoter region in male individuals with aSAH but no significant association with outcome. Another study found suggestive associations between DNAm trajectories within CSF and modified Rankin scale (mRS) at 3 and 12 months, specifically in CpG sites within the hepcidin antimicrobial peptide gene (*HAMP*) [47], a key regulator of brain iron homeostasis. Finally, one study identified a specific CpG site within the *STEAP3* metalloreductase gene associated with unfavorable outcomes [48], emphasizing the gene’s significance in iron toxicity and its implications for aSAH prognosis [49].

The potential of DNAm profiles from blood samples as a reliable surrogate for CSF in aSAH has also been explored [50]. The findings indicate a low correlation between both tissues, suggesting that CSF may be a more suitable source for detecting specific DNAm profiles in cases of aSAH.

### 3.2. Non-Coding RNA

#### 3.2.1. MicroRNA

Multiple studies have demonstrated the dysregulation of miRNAs in patients with both IA and aSAH [23,51,52,53,54,55,56,57,58,59,60,61,62,63,64,65,66,67,68,69,70,71,72,73,74,75,76,77,78,79,80,81,82,83,84] (Table 1). However, it is essential to note that the quality of evidence in most of these studies is generally low. This is due to substantial methodological heterogeneity across the studies, encompassing variations in sample types (plasma, CSF, BA tissue), collection timepoints (ranging from the first day to up to 2 years post bleeding), and laboratory techniques (quantitative polymerase chain reaction [qPCR], next-generation sequencing [NGS], microarrays, NanoString). Furthermore, some studies focused on aSAH patients, others examined unruptured IAs, and some included both ruptured and unruptured IAs. Additionally, most studies have limitations regarding statistical analysis, such as not adjusting for potential confounding variables (such as sex, age, or smoking) or correcting the *p*-values for multiple testing. Significantly, most of these investigations have been conducted in cohorts of Asian descent, with a notable lack of research on other ethnicities or on multiethnic cohorts. Among the 35 studies listed in the table, only nine of them replicated their findings in different cohorts [54,55,56,59,69,70,71,81,85], with just six of them having a total sample size exceeding 50 cases [54,56,59,69,81,85].

Given these limitations, the results exhibit high heterogeneity across studies, with variations in the miRNAs identified and inconsistencies such as miRNA expression being reported in opposite directions. To mitigate this heterogeneity, we checked which miRNAs consistently differentially expressed in the same direction (either upregulated or downregulated) across at least two studies with similar designs. Our analysis identified 18 noteworthy miRNAs associated with IA formation and rupture: let-7b-5p, miR-1297, miR-132, miR-143, miR-145, miR-15a-5p, miR-15b-5p, miR-17-5p, miR-19b-3p, miR-20a-5p, miR-23b-3p, miR-24-3p, miR-26a, miR-27b-3p, miR-29a-3p, miR-324-3p, miR-34c-5p, miR-502-5p and miR-9 (Table 2). Among them, miR-145 and miR-143 stand out, since they are downregulated in different sample types (plasma, serum, and IA tissue) in both ruptured and unruptured IA in six different studies [55,62,66,75,77,84], although a small study reported regulation in the opposite direction in plasma in ruptured IA [60]. These two miRNAs are highly correlated and participate in VSMC migration and proliferation processes [62,66], suggesting that a lower expression of miR-143/145 is potentially associated with IA formation.

Among the studies incorporating bioinformatics pathway analyses, the most frequently associated pathways are related to inflammation, transforming growth factor-beta, mitogen-activated protein kinase (MAPK) signaling, smooth muscle regulation, and extracellular matrix dynamics.

The available studies concerning complications and prognosis following aSAH remain limited. Only a few studies have explored the association between miRNA expression and variables such as VS, DCI, or overall prognosis, and unfortunately, many of these findings lack reproducibility across different cohorts.

In terms of VS, one study identified the downregulation of two miRNAs, miR-125b-5p and miR-143-3p, in patients with worse neurological status and experiencing VS [55]. Notably, miR-143 is highly expressed in VSMC, endothelial cells, and inflammatory cells [86], while miR-125b is associated with cell proliferation, apoptosis, and vascular smooth cell phenotyping in aneurysms [82,87]. However, another study utilizing NGS from peripheral blood found no significant differences between the 14 VS cases and 13 non-VS cases [63].

On the other hand, another study found elevated blood levels of miR-3177-3p in cases with VS, leading to a reduced expression of the Lactate Dehydrogenase A gene (*LDHA*) [88], identified in brain microvascular endothelial cells following hypoxia signaling. Additionally, a smaller study involving 31 aSAH patients reported that miR-142-3p and miR-1274b levels in CSF were promising predictors of VS at day 3 [60]. Other investigations into the CSF miRNA profile in aSAH cases revealed several downregulated miRNAs associated with VS, including miR-27a-3p, miR-516a-5p, miR-566, miR-508-3p, miR-519-3p, miR-337-5p, miR-1197, miR-132-3p, miR-19b-3p, miR-210-3p, miR-221-3p, and miR-484 [72,73], although direct comparisons between studies are challenging due to methodological differences including variations in CSF sampling timing, miRNA detection methods, and VS definition.

Regarding DCI, a study comparing aSAH patients with and without DCI identified a combination of four miRNAs (miR-4463, miR-4532, miR-4793, and miR-1290) that effectively differentiated between the two groups [68]. Another small study analyzed miRNA profiles in CSF from 27 aSAH patients, reporting a relative increase in miR-21-5p and miR-221-3p in aSAH patients with DCI compared to those without [72]. Additionally, miR-221-3p was associated with VS. However, most associations did not remain significant after correction for multiple testing, underscoring the need for replication in confirmatory cohorts.

Concerning the prognosis of aSAH patients, the downregulation of miR-146a-5p and miR-27b-3p in plasma has been associated with better recovery outcomes [56], while lower serum levels of miR-502-5p and miR-1297 have been observed in patients with poor outcomes [64,65,69]. Additionally, increased levels of miR-9-3-p and miR-9-5-p in CSF samples have been linked to poor functional outcomes [85]. Other studies in ischemic stroke have also associated miR-9 in CSF or plasma with higher severity and infarct volume, making it a promising candidate for future investigations.

#### 3.2.2. Long Non-Coding RNA

Research on lncRNA regarding IAs and aSAH is still limited. Of the six studies we found, two of them analyzed peripheral blood or plasma and four analyzed IA tissues [89,90,91,92,93,94]. Characteristics and main findings of the studies are summarized in Table 3. Most of them quantified lncRNA and mRNA to construct co-expression networks in order to predict possible pathways implicated in IA formation. Due to their methodological differences, the results are not completely comparable, but the majority of them found an association with pathways related to inflammatory response or muscle tissue development and contraction. Only one of the studies replicated its results in an independent cohort [90], and five of the six studies have a sample size lower than 50 cases.

Regarding complications and outcomes, our investigation identified relevant findings exclusively from a singular study [91]. In this study, they employed RT-qPCR to quantify the expression of Metastasis-Associated Lung Adenocarcinoma Transcript 1 (MALAT-1) in plasma samples from 105 patients with IA, consisting of 69 cases with rupture and 36 without rupture. Additionally, 40 control subjects were included in the study. The results demonstrated a significant upregulation of MALAT-1 in IA patients, and its elevated levels were associated with both the rupture of the aneurysms and unfavorable clinical outcomes. MALAT-1 has been implicated in the pathogenesis of various conditions, including IA, aortic aneurysms, ischemic stroke, myocardial acute infarction, and vascular remodeling in hypertension [95]. Given these diverse associations, the identification of MALAT-1 as a potential therapeutic target in IA becomes particularly interesting. However, it is imperative that these promising findings are validated in further studies.

#### 3.2.3. Circular RNA

We only found four studies that explored the relationship between circRNA and IA, all limited either by sample size or lack of replication. These studies suggest the involvement of certain circRNAs in pathways such as the mammalian target of rapamycin (mTOR) [96] or pathways related to the functions of VSMCs in the development of IA [97]. Additionally, they indicate associations with pathways related to leukocyte migration in patients with aSAH [98]. A larger study including 216 IA patients and 186 healthy controls explored the potential of hsa_circ_0000690 as a biomarker for aSAH diagnosis and prognosis, as it was described to be involved in aSAH in a previous study [98,99]. They found a lower expression in IA patients compared to the controls and the correlation with aSAH severity and bleeding volume. Again, inflammation and leukocyte function was found to be involved in aSAH.

#### 3.2.4. Studies Based on Bioinformatic Analysis: Regulation of Gene Expression through Non-Coding RNA

Several studies identified in the literature leveraged data from the Gene Expression Omnibus (GEO) database to conduct bioinformatic analyses, integrating information on miRNA, lncRNA, mRNA, and proteins. This enabled the construction of co-regulatory expression networks. The GEO database, an international public repository, archives and freely shares high-throughput gene expression and functional genomics datasets [100]. These datasets stem from previous studies, and the purpose of the bioinformatic analysis is to predict interactions among different types of RNAs. We found six studies of this kind specifically focused on IA and aSAH. A primary limitation in these studies remains the small sample sizes within the GEO datasets explored. However, it is noteworthy that some studies have validated their findings in distinct GEO datasets [101,102,103].

Within this body of research, certain ncRNA and pathways have been proposed to play crucial roles in IA formation and rupture. Notable examples include miRNA-125b and its target gene nitric oxide synthase 1 (NOS1) [104]. Additionally, numerous differentially expressed genes associated with immune response [102,103,104], as well as pathways such as cell proliferation and the PI3K/Akt signaling pathway [101,105], Nuclear Factor Kappa B Subunit 1, toll-like receptor, and sphingolipid signaling pathways [106] have been implicated.

Furthermore, miR-191-3p, miR-423-5p, miR-424-5p, and miR-425-3p have demonstrated significant potential in predicting IA occurrence [101], and a comprehensive set of 38 RNA signatures, comprising two lncRNAs (JMJD1C-AS1 and LINC01144), one microRNA (miR-510), and 35 mRNAs, has been employed to construct a classifier that exhibited promising predictive capabilities for assessing the risk of aSAH [102].

### 3.3. Histone Modification

Limited studies have investigated histone modifications in relation to IA risk. One study utilized bioinformatics to analyze specific histone marks, H3K4me1 and H3K27ac, and transcription-factor binding sites associated with IA across various cell types [107]. Human umbilical vein endothelial cells (HUVECs) showed significantly higher histone marks in genomic regions linked to IA risk, emphasizing the potential role of endothelial tissue. Another study used ChIP-seq technology on human Circle of Willis tissue, identifying histone H3K4me1 and H3K27ac modifications in regulatory regions overlapping with known IA-associated single nucleotide polymorphism (SNP) regions [108]. These findings suggest a connection between IA-associated SNPs and arterial tissue regulatory elements. Bioinformatics analysis revealed the enrichment of processes related to vasculature development, extracellular matrix, and cell adhesion in these regulatory regions, implicating their significance in IA pathogenesis.

Figure 1 provides a summary of the main epigenetic mechanisms involved in IBA and aSAH pathophysiology.

## 4. Discussion

Our review of the epigenetic influence of IA and aSAH pathophysiology revealed critical insights into a field that has gained interest in recent years. Our findings align with the growing recognition that genetic predisposition, while significant, does not fully account for disease heritability [5]. This gap suggests a potential pivotal role for epigenetic modifications influenced by environmental factors, notably in the context of key risk factors such as age, smoking, and hypertension, which are known to affect both the epigenome and IA risk [109,110].

### 4.1. Epigenetic Mechanisms and Implications

The studies on DNAm in relation to IA and aSAH, although limited in number, provide valuable insights into the epigenetic regulation of genes potentially involved in the pathogenesis of these conditions (Figure 1). The differential methylation observed in promoter regions of genes like *NOS1AP*, *PDGFD* and *MAP3K10* offers a promising avenue for understanding the molecular mechanisms underlying IA formation [37,38,39]. Furthermore, hypermethylation at *INSR* and *CDHR5* genes in patients with DCI underscores the potential of DNAm as a prognostic marker [42]. However, these results are mitigated so far by methodological limitations, such as small sample sizes and the absence of replication in diverse populations.

The role of miRNAs in IA and aSAH has been more extensively studied, but again is complicated to have solid evidence due to methodological heterogeneity and inconsistent findings. Despite this, the identification of dysregulated miRNAs in biofluids and aneurysmal tissues points towards their involvement in key pathophysiological processes such as VSMC regulation, endothelial function, and inflammatory response. Notably, the consistent dysregulation of specific miRNAs, like miR-145 and miR-143, across studies [55,62,66,75,77,84] suggests a potentially significant role in IA and aSAH pathogenesis.

Research on lncRNAs and circRNAs is still emerging. Early studies have linked lncRNAs to pathways involved in inflammation and muscle tissue development, which are crucial in aneurysm formation and rupture [89,90,91,92,93,94]. However, the small sample sizes and lack of replication limit the current understanding of their roles. circRNAs have shown potential links to IA through bioinformatic analyses. Their involvement in pathways like the mTOR signaling pathway points towards a possible role in the pathogenesis of IA [96]. Nonetheless, the research in this field is nascent, and much remains to be explored regarding the specific functions and mechanisms of circRNAs in IA and aSAH.

Although scarce, the current research focused on histone modifications reveals a potential link between epigenetic regulation at the level of chromatin structure and IA risk. Studies utilizing ChIP-seq technology have identified specific histone marks associated with IA-associated SNP regions, particularly in arterial tissue, suggesting a direct connection between genetic risk factors and epigenetic regulation [108]. However, the characterization of histone modifications in the field of IA and aSAH is still in a very early stage, and we will have to wait for future studies to truly understand the role that this epigenetic modification plays in the pathophysiology of this disease.

### 4.2. Limitations

This narrative review has several limitations: The exclusive use of PubMed and Embase, and limiting our review to English-language articles, might have led to the exclusion of significant studies in other databases or languages. By omitting animal, in vitro, genetic, and transcriptomic studies, we may have overlooked comprehensive insights into the epigenetic mechanisms of IA and aSAH. As a narrative, rather than a systematic review or meta-analysis, our conclusions are based on descriptive synthesis, lacking the statistical power of more rigorous methodologies. These limitations should be considered in the context of the review’s findings.

### 4.3. Future Directions

In light of what has been shown in this review and regarding the methodological limitations identified, we propose a series of recommendations for future research. These recommendations are designed to further the progress in epigenetics research, addressing and transcending the constraints of prior investigations.

In the field of DNAm, larger-scale EWAS are needed that can provide sufficient discovery power, and in turn allow for the necessary statistical adjustments to be made to achieve meaningful results. With the rapid advancements in high-throughput sequencing and bioinformatics tools, EWAS now have the potential to analyze DNAm changes with greater resolution and accuracy than ever before. This technological evolution allows for the identification of novel epigenetic biomarkers that could serve as targets for therapeutic intervention or as tools for early disease detection and prognosis. Furthermore, integrating the findings of EWAS with other omics data, such as genomics, transcriptomics, and proteomics, can enrich our understanding of the molecular mechanisms underlying IA and aSAH. In this direction, future studies should analyze the interplay between circRNA and miRNA. Such integrative approaches can help in identifying key molecular pathways disrupted in these conditions and pave the way for the development of targeted therapies.

Secondly, investigating methylation quantitative trait loci (mQTLs) may help in better understanding the interplay between genetic predisposition and environmental factors. mQTLs analysis involves examining specific genetic loci that influence the methylation state of DNA, thereby impacting gene expression and potentially, disease phenotypes [111]. Identifying these loci can reveal the genetic contribution to the susceptibility to environmental factors, such as smoking or hypertension, which contribute to risk and progression of IA or aSAH. Such insights are not only pivotal for understanding disease etiology but also have significant implications for personalized medicine. Given the severe consequences of aSAH, particularly in young women, and the absence of efficacious preventative interventions, identifying novel therapeutic targets to mitigate the development and rupture of IA, as well as associated complications, is of critical importance. Recent progress in the field of epigenetics presents a promising avenue for the exploration of innovative therapeutic strategies. Epigenetic modifications, implicated in the pathogenesis of IA, may be reversible. This reversibility provides a foundation for interventions that could potentially alter these epigenetic changes. Such a strategy enables the development of patient-specific treatments, aligned with individual epigenetic profiles, paving the way for more personalized tailored medicine.

Finally, understanding the temporal relationship between epigenetic modifications and the onset and progression of IA and aSAH is a crucial next step in this field. Future research should prioritize longitudinal studies tracking epigenetic changes over time in large population cohorts, ideally from a pre-disease state through to disease development and progression. Such studies are invaluable in discerning the temporal sequence and potential causal role of epigenetic alterations in these cerebrovascular conditions. Notably, ongoing population studies, such as the Genomes for Life (GCAT) project [112], promise to shed light on these dynamics. Concurrently, Mendelian randomization studies could present a complementary approach. These studies could employ genetic variants as instrumental variables to establish causal links between specific epigenetic markers and disease outcomes.

## 5. Materials and Methods

Due to the scarcity of studies available in this field, as well as their diverse nature and methodological constraints, we chose to perform a narrative review. We searched for manuscripts published between January 2000 and June 2023, focusing on the role of epigenetic mechanisms in IA and aSAH. This time period was selected due to the significant advancement in epigenetic research after 2000, as earlier studies were limited in scope and frequency due to technological constraints. We used the PubMed and Embase databases, targeting articles containing at least one key term related to IA and aSAH (including ‘brain aneurysm’, ‘intracranial aneurysm’, ‘cerebral aneurysm’, ‘subarachnoid hemorrhage’, ‘outcome’, ‘delayed cerebral ischemia’, ‘vasospasm’) and one or more key terms related to epigenetics (‘epigenetic’, ‘methylation’, ‘EWAS’, ‘microRNA’, ‘long-non-coding RNA’, ‘histone modification’, ‘circular RNA’). We additionally excluded studies that were conducted using animal or in vitro models, those focused primarily on genetics or transcriptomics, and reviews. Additionally, we limited our review to articles published in English. The screening of articles was meticulously conducted by I.F.P, A.M.G, and A.S.P. For each selected paper, we extracted comprehensive information including the epigenetic mechanisms analyzed, study design, country of origin, sample types and collection methods, analytical methodologies, outcomes, main results, and limitations. These data were synthesized and presented in a tabular format for ease of comparison and analysis. The review of extracted information and the drafting of the main conclusions were collaboratively undertaken by I.F.P, J.J.B, and E.C.G. Following our systematic approach in reviewing the literature, we have organized our findings into distinct sections, based on the different epigenetic mechanisms and their role in IA formation and rupture, aSAH complications, and aSAH prognosis. This structure allows us to present a clear and comprehensive overview of the current state of research in each area, highlighting the most significant discoveries and identifying areas where further study is needed.

## 6. Conclusions

Our comprehensive review highlights a key moment in the field of epigenetics concerning IA and aSAH. Key findings from the reviewed studies suggest that differential DNAm in genes such as NOS1AP, PDGFD and MAP3K10, INSR, and CDHR5 may contribute to IA formation and the development of DCI, respectively. Moreover, research on miRNAs indicates their involvement in regulating VSMCs, endothelial function, and inflammation. Notably, the consistent dysregulation of miR-145 and miR-143 across studies suggests their significant role in IA and aneurysmal aSAH pathogenesis. Emerging studies on lncRNAs and circRNAs link them to inflammation and pathways related to muscle tissue development, which are relevant to aneurysm formation. Additionally, initial findings on histone modifications suggest a potential connection between genetic risk factors for IA and specific histone marks, such as H3K4me1 and H3K27ac.

While the current body of research provides valuable insights into the potential roles of various epigenetic mechanisms, it also underscores the need for more comprehensive and methodologically robust studies. Addressing the current limitations and expanding the scope of research will be pivotal in elucidating the underlying mechanisms and translating these findings into clinical applications. The potential for personalized therapeutic strategies targeting epigenetic modifications presents an exciting frontier in the management of IA and aSAH, offering hope for improved outcomes in these complex cerebrovascular disorders.

## Figures and Tables

**Figure 1 ijms-25-03433-f001:**
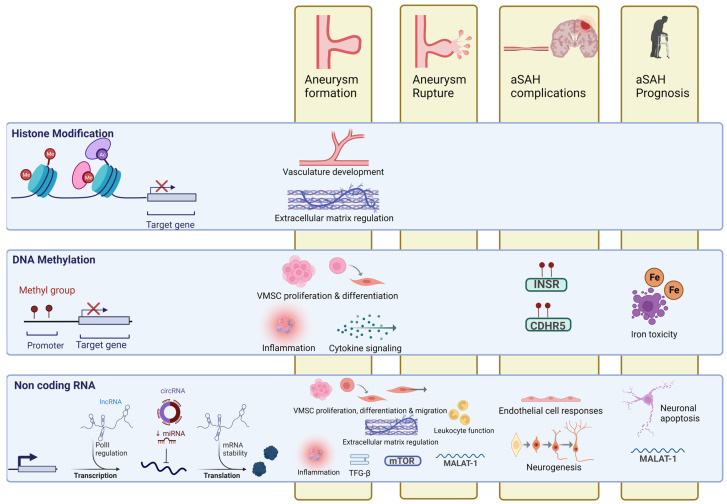
Role of the different epigenetic mechanisms in intracranial aneurysm formation and rupture, as well as their impact on complications and prognosis following aSAH. Regarding aneurysm formation and rupture, the most frequently reported pathways involve VSMCs, immune response, and inflammation. Hypermethylated CpGs in the INSR and CDHR5 genes have been associated with delayed cerebral ischemia (DCI), while miRNAs implicated in endothelial cell responses and neurogenesis have been linked to vasospasm and DCI. Differential DNAm in certain genes associated with iron toxicity, miRNAs involved in neuronal apoptosis, and the lncRNA MALAT-1 have been reported to influence the prognosis of aSAH. Keywords: aSAH: aneurysmal subarachnoid hemorrhage; CDHR5: cadherin-related family member 5; circRNA: circular RNA; INSR: insulin receptor; lncRNA: long non-coding RNA; MALAT-1: metastasis-associated lung adenocarcinoma transcript 1; miRNA: microRNA; mRNA: messenger RNA; mTOR: mammalian target of rapamycin; VMSC: vascular muscle smooth cell. Created with Biorender.com.

**Table 1 ijms-25-03433-t001:** Studies evaluating miRNA expression in brain aneurysm (BA) and aneurysmal subarachnoid hemorrhage (aSAH).

Reference	Cohorts	Sample and Collection Day	Detection Method	DE miRNAs	Pathway/Functional Analysis	Other Findings
[51] China	25 uIA/20 controls	Serum	RT-qPCR	miR-140 upregulated in IA	Apoptosis and inhibition of proliferation of HVSMCs	
[52] Turkey	50 IA (34 rIA and 66 uIA)/50 controls	IA tissue Control: STA	RT-qPCR	miR-26a, miR-29a, and miR-448-3p upregulated in IA	Extracellular matrix regulation, HVSMCs proliferation and apoptosis, oxidative stress	
[53] China	69 rIA and 66 uIA, 68 hydrocephalus patients as controls	CSF at time of admission	RT-qPCR	miR-152-3p downregulated in IA	Proliferation and migration of HVSMCs	Lower miR-152-3p levels in rIA patients compared to uIA
[54] China	Screening: 5 aSAH/5 controls. Validation in 2 cohorts: - 20 aSAH/40 controls - 40 aSAH/30 controls	Plasma. Day 1	Screening: microarray. Validation: RT-qPCR	Screening 14 DE miRNAs. Validation: miR-23b-3p, miR-590-5p, miR-20b-5p, miR-142-3p, miR-29b-3p, all downregulated in aSAH	Inflammation, smooth muscle cell proliferation, and cell adhesion. TGF-beta pathway	
[23] China	31 aSAH (14 with VS)/ 8 controls	CSF and plasma collected at 3, 5, 7, and 10 days	RT-qPCR	Let-7b-5p, miR-15b-5p, miR-17-5p, miR-19b-3p, miR-20a-5p, miR-24-3p, and miR-29a-3p, all upregulated in aSAH in CSF on days 3 and 7.	Angiogenesis, inflammatory and immune response, proliferation and apoptosis, oxidative stress, mitochondrial function, neurogenesis, and differentiation	A panel including 26 miRNAs on CSF at day 3 had an AUC of 0.8 for VS prediction.
[55] India	Screening: 8 aSAH/8 controls. Validation: 29 aSAH/20 controls ^1^	IA tissue Control: Intercostal arteries	Screening: microarray. Validation: RT-qPCR	Screening: 70 miRNAs DE (67 downregulated in aSAH). 10 selected miRNAs are validated: miR-24-3p, miR-26b-5p, miR-27b-3p, miR-125b-5p, miR-143-3p, miR-145-5p, miR-193a-3p, miR-199a-5p, miR-365a-3p/365b-3p, miR-497-5p	TGF- beta and MAPK pathways. Inflammation, extracellular matrix and VSMC degradation and apoptosis	miR-125b-5p, miR-143-3p, miR-199a-5p decreased in patients with WFNS 3-4. miR-125b-5p, miR-143-3p, decreased in VS
[56] India	Screening: 20 aSAH/20 controls. Validation: 88 aSAH ^1^/110 controls and 25 spontaneous non-aneurysmatic SAH	Plasma. Day 1	Screening: RT-qPCR (179 miRNAs panel). Validation: RT-qPCR	Screening: DE 76 miRNAs (35 upregulated and 41 downregulated in SAH). 8 Selected miR validated: miR-15a-5p, miR-34a-5p, miR-374a-5p, miR-146a-5p, miR-376c-3p, miR-18b-5p, miR-24-3p, miR-27b-3p	Inflammation. TFG-beta, MAPK, focal adhesion, PI3K/Akt pathways	Expression patterns of the 8 miR in the non-aneurysmal SAH cohort are similar to controls. miR-146a-5p and miR-27b-3p associated with outcome
[57] USA	19 IPH/17 aSAH/21 IS (no controls)	Extracellular vesicles from plasma. Day 1	NGS	ex-miRNA that distinguished aSAH from IPH: 68. aSAH from IS: 52		A 25 miR panel can discriminate aSAH from other stroke subtypes (AUC 0.927)
[58] China	102 IA patients (79 rIA, 23 uIA)/ 80 controls	Serum extracted before treatment	RT-qPCR	miR-126 upregulated in IA	MAPK signaling pathway	miR-126 was an independent risk factor for IA rupture
[59] China	Screening: 4 uIA/8 rIA/4 controls. Validation: 30 uIA/39 rIA/30 controls ^1^	Plasma exosomes extracted within the first 7 days in aSAH	Screening: NGS. Validation: RT-qPCR	Screening: - 29 miRNAs DE between uIA and controls - 31 miRNAs DE between rIA and controls - 121 DE miRNAs between uIA and rIA. Validation (from 5 selected miRNAs): miR-145-5p and miR-29a-3p upregulated in uIA and rIA vs. controls		
[60] Poland	19 acute aSAH (first 72 h)/20 chronic aSAH (3–15 months)/20 controls	Peripheral blood (before neurosurgical intervention)	NGS	196 miRNAs DE between 3 groups. 2 patterns: - 81 miRNAs DE in the acute phase that return to normal levels in the chronic phase - 11 miRNAs downregulated in the chronic phase	Cytokine-cytokine receptor interactions. Control of immune cell homeostasis	
[61] China	48 rIA/46 uIA	IA tissue and peripheral blood ^2^	qPCR	miR-155 upregulated in uIA vs. rIA	miR-155 downregulates matrix metalloproteinase-2, implicated in extracellular matrix degradation	SNP rs767649 could inhibit miR-155 transcription
[62] China	30 IA ^3^/30 controls	Serum	qPCR	miR-145 and miR-143 are decreased in IA patients	Suppress VSMCs proliferation and migration with upregulation of Krüppel-like factor 5	
[63] Brazil	27 aSAH (14 with VS)/6 controls	Peripheral blood. Day 7–10	NGS	5 miRNAs downregulated (let-7f-5p, miR-486-5p, miR-126-5p, miR-17-5p, miR-451a), and 3 upregulated (miR-146a-5p, miR-589-5p, miR-941)	Proto-oncogenes, caspase activation and apoptosis, cyclin kinase regulators, growth factors	miR-486-5p associated with poor outcome
[64] China	128 aSAH/40 controls	Serum. 24 h, 72 h, day 7, day 14.	RT-qPCR	miR-1297 upregulated in aSAH since day 1, reaching a peak at 7 days		miR-1297 at 24 h negatively correlated with WFNS grade, miR-1297 at 24 and 72 h associated with mRS scale at 1 year
[65] China	129 aSAH/40 controls	Serum. 24 h, 72 h, day 7, day 14.	RT-qPCR	miR-502-5p upregulated in aSAH since day 1, reaching a peak at 7 days		miR-502-5p at 3 and 7 days negatively correlated with WFNS grade, miR-502-5p at days 1,3 and 7 associated with mRS scale at 1 year
[66] China	9 uIA/8 rIA/7 controls	Plasma. Day 1-3	RT-qPCR	miR-143 and 145 downregulated in rIA vs. controls. Both miR were correlated	Vascular inflammation	rIA patients had higher levels of matrix-metalloproteinase-9 than uIA and controls
[67] China	32 uIA/17 controls	IA tissue vs. VSMCs isolated from vessel walls	RT-qPCR	miR-23b-3p downregulated in IA.	miR-23b-3p targets phosphatase and tensin homolog, interfering with viability and apoptosis of VSMCs	
[68] China	40 aSAH (20 with DCI)/20 controls	Plasma. Day 7	RT-qPCR	The combination of 4 miRNAs (miR-4532, miR-4463, miR-1290 and miR-4793) could distinguish aSAH patients from controls	Developmental pathways as Wnt signaling pathway, hedgehog and oxytocin signaling pathways	The combination of the same 4 miR could distinguish patients with and without DCI.
[69] China	Screening: 3 aSAH/3 controls. Validation: 60 aSAH/10 controls	Serum. Day 3	Screening: microarray. Validation: RT-qPCR	Screening: 13 miRNAs upregulated. Validation: miR-502-5p, miR-1297, miR-4320 upregulated in aSAH		miR-1297 and miR-502-5p were negatively correlated with WFNS grade and associated with poor outcome
[70] Netherlands	Screening: 15 prior aSAH (at least 2 years before, 11 with and additional uIA)/15 controls. Validation: 15 prior aSAH/15 uIA/15 controls	Plasma	Screening: RT-qPCR. Validation: RT-qPCR	Screening: - 3 DE miRNAs in aSAH - 2 DE miRNAs in aSAH with additional uIA vs. controls. Validation: - miR-183-5p downregulated in aSAH and uIA vs. controls - miR-200a-3p upregulated in aSAH vs. controls - Let-7b-5p downregulated in uBA vs. controls	miR-183-5p, miR-200a-3p and let-7b-5p regulated 15 genes previously described to be involved in IA development and rupture, implicated in cell proliferation, extracellular matrix composition and inflammation	
[71] Japan	Screening: 2 aSAH/2 controls Validation: 8 aSAH/3 controls ^1^	Plasma and CSF Screening: day 3. Validation: days 1, 3, 5, 7, 9, 11, 13	Screening: Microarray. Validation: RT-qPCR	Screening: - Plasma: 19 miRNAs DE - CSF: 42 miRNAs DE. Validation (2 selected miRNAs): - miR-6724 downregulated in plasma and CSF from day 1 to 3. - miR-15a: upregulated in plasma on days 5 and 7; and upregulated in CSF on days 1,3, and 5		
[72] Denmark	27 aSAH with EVD/10 controls. Screening and validation in the same cohort	CSF within the first 5 days	RT-qPCR. 2 different platforms	Screening: 151 miRNAs DE. Validation: 66 upregulated miRNAs in aSAH		
[73] Australia	20 aSAH (10 with VS) with EVD/4 controls	CSF extracted between day 1 and 8	NanoString	33 miRNAs DE between different aSAH groups (VS or not, day 1 samples or all) and controls. miR-451a upregulated in almost all aSAH groups		6 miRNAs DE in VS
[74] China	8 IA ^3^ vs. 3 controls	IA tissue Control: ‘normal cerebral arteries’	qPCR	miR-370-3p increased in IA.	Cell proliferation and angiogenesis	
[75] (China-Han population)	62 IA (rIA and uIA mixed)/62 controls	Plasma ^2^	qPCR	miR-143 and 145 were downregulated in IA		rs4705342 TC and TC/CC genotypes in the promoter of the miR-143/145 cluster were related to a lower risk of IA
[76] China	13 aSAH/11 controls	IA tissue Control: Middle meningeal artery	qPCR	miR-9 increased in aSAH	miR-9 interferes with the viability and contractility of VSMC by targeting myocardin	
[77] USA	7 uIA/10 controls	IA tissue Control: STA	Microarray. NanoString for technical validation	19 miRNAs upregulated and 5 downregulated in uIA. Strongest changes in: miR-21, miR-143-5p and miR-145	Collagen formation, inflammation regulation, lipid metabolism, smooth muscle phenotypic modification and extracellular matrix remodeling	They also found 1028 genes DE and investigated miRNA-mRNA pairs
[78] USA	8 aSAH women with EVD (all Fisher III). No controls	CSF daily collected during days 3 to 12	NanoString RT-qPCR for technical validation in 3 selected miRNAs	52 miRNAs detected divided in 2 clusters: one with decreased abundance over time and another with increasing abundance over time		
[79] China	Screening: 3 aSAH/3 controls. Validation: 70 aSAH ^1^/10 controls	IA tissue Control: Middle meningeal artery	Screening: microarray. Validation: RT-qPCR	Screening 17 DE miRNAs Validation: miR-34c-5p, miR-539-5p, miR-431-5p, miR-3651, and miR-758-3p, all upregulated en aSAH	VSMCs differentiation by targeting myocardin	
[80] China	Screening: 20 aSAH with DCI /20 controls. Validation: 20 aSAH with DCI ^1^ + 20 aSAH without DCI /20 controls	Plasma. Day 7	Screening: microarray. Validation: qPCR	Screening: 99 DE miRNAs. Validation: miR-132 and miR-324 upregulated in both SAH DCI and non-DCI groups vs. controls		
[81] China	Screening: 40 IA (20 uIA and 20 rIA)/20 controls. Validation: 93 IA ^3^/50 controls. Validation of potential biomarkers: 17 controls and 26 IA randomly selected from a combination of the 2 previous cohorts	Plasma	Screening: microarray. Validation: microarray. Validation of potential biomarkers: RT-qPCR	Screening: 119 miRNAs DE in uIA, 23 in rIA and 20 in both vs. controls. 73 miRNAs DE in uIA vs. rIA. Validation: 99 DE miRNAs in IA vs. controls. Potential biomarkers: miR-16 and miR-25 upregulated in IA	Inflammation and connective tissue	miR-16 and miR-25 are good predictors of IA
[82] China	6 rIA/6 controls. Validation: ‘semi-independent sample including new and the original samples’	IA tissue Control: STA	Microarray RT-qPCR for technical validation in 3 miR	157 DE miRNAs in IA. Technical validation confirms that miR-99b and miR-493 were upregulated and miR-340 downregulated	Endothelium and vascular smooth muscle-enriched miR. Inflammation, dysregulation of extracellular matrix, smooth muscle cell proliferation, programmed cell death and response to oxidative stress. Protein translation process	
[83] China	4 groups: A: 6 uIA with daughter aneurysms B: 6 uIA without daughter aneurysms C: 6 rIA D: 6 controls	Plasma ^2^	Microarray	86 DE miRNAs between any of the IA group and controls	Apoptosis and activation of cells associated with function of vascular wall	
[84] China	14 rIA/14 controls Screening and validation in the same cohort.	IA tissue Control: middle meningeal artery	Screening: Microarray. Validation: RT-qPCR	Screening: 30 DE miRNAs Validation: 18 DE miRNAs, all downregulated	Migration of phagocytes, proliferation and cell movement of mononuclear leukocytes, cell movement of smooth muscle cells	

^1^ The validation cohort also includes the samples used in the screening phase. ^2^ The study does not specify the moment of sample collection. ^3^ The study does not specify if IA are ruptured or not. Keywords: CSF: Cerebrospinal Fluid; DE: differentially expressed; DCI: delayed cerebral ischemia; EVD: extraventricular drainage; ex-miR: exosomal microRNA; HVSMC: human vascular smooth muscle cell; IA: intracranial aneurysm; IPH: intraparenchymal hemorrhage; IS: ischemic stroke; MAPK: mitogen-activated protein kinases; miR: microRNA; mRS: modified Rankin scale; NGS: Next generation sequencing; rIA: ruptured intracranial aneurysm; RT-qPCR: Reverse transcription quantitative polymerase chain reaction; SNP: Single nucleotide polymorphism; STA: Superficial temporal artery; TGF: Transforming growth factor; uIA: unruptured intracranial aneurysm; VS: vasospasm; WFNS: World Federation of Neurosurgical Societies.

**Table 2 ijms-25-03433-t002:** MicroRNAs consistently differentially expressed across at least two of the reviewed studies with similar designs. Keywords: CSF: cerebrospinal fluid; IA: intracranial aneurysm.

microRNA	Studies	Sample Type	Expression
let-7b-5p	[23,72]	CSF	Upregulated in ruptured IA
miR-1297	[64,69]	Serum	Upregulated in ruptured IA
miR-132	[72,80]	CSF Plasma	Upregulated in ruptured IA
miR-143/miR-145	[55,62,66,72,75,77,80,84]	Plasma Serum IA tissue	Downregulated in ruptured and unruptured IA
miR-15a-5p	[56,72]	Plasma CSF	Upregulated in ruptured IA
miR-15b-5p	[23,72]	CSF	Upregulated in ruptured IA
miR-17-5p	[23,72]	CSF	Upregulated in ruptured IA
miR-19b-3p	[23,72]	CSF	Upregulated in ruptured IA
miR-20a-5p	[23,72]	CSF	Upregulated in ruptured IA
miR-23b-3p	[54,67,84]	Plasma IA tissue	Downregulated in ruptured and unruptured IA
miR-24-3p	[23,72]	CSF	Upregulated in ruptured IA
miR-26a	[52,72]	IA tissue CSF	Upregulated in ruptured and unruptured IA
miR-27b-3p	[55,56]	IA tissue Plasma	Downregulated in ruptured IA
miR-29a-3p	[23,52,59,72]	Plasma CSF IA tissue	Upregulated in ruptured and unruptured IA
miR-324-3p	[72,81]	CSF Plasma	Upregulated in ruptured and unruptured IA
miR-34c-5p	[72,79]	CSF IA tissue	Upregulated in ruptured IA
miR-502-5p	[65,69]	Plasma	Upregulated in ruptured IA
miR-9	[76,85]	CSF IA tissue	Upregulated in ruptured IA

**Table 3 ijms-25-03433-t003:** Studies about the role of long non-coding RNA (lncRNA) in brain aneurysm (IA) formation and aneurysmal subarachnoid hemorrhage (aSAH).

Reference	Cohorts	Sample and Collection Day	Detection Method	DE lncRNA and mRNA	Pathway Analysis	Other Findings
[89] China	20 IA/20 STA	IA tissue	RT-qPCR	lncRNA Antisense Non-coding RNA in the INK4 Locus (ANRIL) enhances VSMC proliferation through mir-7/FGF2 pathway	VSMC proliferation and suppresses apoptosis	
[90] China	Screening: 5 aSAH/5 controls. Validation: 30 aSAH/20 controls	Plasma	Screening: Microarray. Validation: RT-qPCR of 4 selected candidates. mRNA microarray for constructing networks	Screening: 797 DE lncRNAs. Validation: TCONS 00000200 and ENST00000511927 upregulated and ENST00000421997 and ENST00000538202 downregulated in aSAH	Co-expression regulatory networks including the 4 validated lncRNAs and 144 mRNA. Pathways related to muscle tissue development and lymphocyte negative regulation	ROC curve suggests that TCONS 00000200 could serve as biomarker of BA
[91] China	105 IA (69 rIA and 36 uIA)/40 controls	Plasma	RT-qPCR	MALAT-1 upregulated in IA		MALAT-1 also associated with hypertension history, IA rupture, HH level and poor prognosis
[92] China	27 IA (12 rIA and 15 uIA)/27 paired controls	IA tissue Control: STAs from the same patients	Microarray	4129 DE lncRNAs and 2926 DE mRNA	Co-expression regulatory networks: 72 lncRNAs and 34 mRNAs implicated in 4 pathways: VSMC contraction, immune response, inflammatory response and muscle contraction	
[93] China	12 IA patients (6 rIA and 6 uIA)/12 paired controls	IA tissue Control: STAs from the same patients	Microarray	1150 DE lncRNAs, 2545 DE mRNAs, and 286 DE miRNAs	ceRNA network consisting of 8401 miRNA-lncRNA-mRNA interactions. Pathways related to muscle contraction and VSMC contraction	
[94] China	12 IA (6 rIA and 6 uIA)/12 paired controls	IA tissue Control: STAs from the same patients	Microarray RT-qPCR in 8 randomly selected lncRNAs and mRNAs for technical validation	1518 DE lncRNAs and 2545 DE mRNA. 4 lncRNAs and 4 mRNAs replicated with RT-qPCR	Co-expression regulatory networks: 559 lncRNAs, 408 mRNAs. Involved in VSMC contraction, immune response and inflammatory response pathways	

Keywords: ceRNA: competing endogenous RNA; DE: differentially expressed; HH: Hunt and Hess; IA: intracranial aneurysm; MALAT-1: metastasis-associated lung adenocarcinoma transcript 1; mRNA: messenger RNA; miRNA: microRNA; rIA: ruptured intracranial aneurysm; RT-qPCR: reverse transcription quantitative polymerase chain reaction; uIA: unruptured intracranial aneurysm; STA: superficial temporal artery; VSMC: vascular smooth muscle cell.

## Data Availability

Data are contained within the article.

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
