# Peer review of "The Role of Epigenetics in Brain Aneurysm and Subarachnoid Hemorrhage: A Comprehensive Review"

_ijms, 2024, doi:10.3390/ijms25063433_

Round 1
Reviewer 1 Report
Comments and Suggestions for Authors
Dear Authors
The manuscript is an interesting and comprehensive review. In conclusion, some general point is written and I expect to see the most important epigenetic changes as the conclusion. I mean which factors are more important in DNA methylation of CpG islands, lncRNA, miRNA, and histone modification?
Reviewer 2 Report
Comments and Suggestions for Authors The authors are presenting a comprehensive review about the role of epigenetics in brain aneurysm and subarachnoid hemorrhage. The authors recognized that one of the major limitations is the narrative nature or the review. I do have the following points that might help improving the quality of the study: 1) Please provide a figure resuming the overview of epigenetics and its mechanisms 2) In the conclusion " Our investigation into the epigenetic influence of BA and aSA". I suggest replacing "our investigation" by "our review". Best regardsAuthor Response
Please see the attachment

Reviewer 3 Report
Comments and Suggestions for Authors
The present review about the relation/implication of epigenetic on BA and aSAH is well written and conducted, on the other hand few problems are encountered as follow:
1) It is not clear if the authors performed a systematic PRISMA review and in case the PRISMA flow chart is missing (please add)
2) the methods section should be introduced after the introduction and before the results sections and few lines about the review process methods should be added.
3) results sections is too long and often redundant, please shorten it as much as possible
4) few lines more should be added about all possible personalized/tailored future therapeutic way to improve globally the outcome of such pathology
Reviewer 4 Report
Comments and Suggestions for Authors
The manuscript entitled ‘The Role of Epigenetics in Brain Aneurysm and Subarachnoid Hemorrhage: A Comprehensive Review’ written by Isabel Fernández-Pérez et al. gathers data regarding the role of DNA methylation, non-coding RNAs, and histone modification in cerebral aneurysm and aneurysmal subarachnoid hemorrhage. The topic discussed in the work is particularly important due to the high mortality rate, severe complications, and significant socio-economic impact of these conditions. Despite many studies, there is still a lack of efficient methods for the diagnosis, prognosis, and therapy of cerebral aneurysmal diseases. The elucidation of the epigenetic mechanisms underlaying these diseases may provide promising targets for their better management. The review prepared by the authors gives a clear and comprehensive image of current progress in this field. The authors described the results of previously performed studies with constructive criticism, extracted the most significant points, indicated numerous limitations, and outlined directions for the future.
The manuscript is well structured, relevant to the field, and has appropriate references and informative tables and figure. The English language is clear.
Specific comments
I have only a few minor remarks to make the text more clear.
1. The medical terms of ‘brain aneurysm’ are ‘intracranial aneurysm’ or ‘cerebral aneurysm’; therefore, one of these medical terms could be used in the manuscript instead of ‘brain aneurysm’.
2. In the text, some unnecessary white spaces (e.g. in lines 69 and 175), as well as lacking white spaces (e.g. in lines 77, 87, 128, 213, and 227) occur. The text layer of the manuscript should be checked once again.
3. In the legend to Figure 1, it could be beneficial for the readers to provide a short description of the illustrated mechanisms.
I believe that my suggestions will help the authors improve the quality of their manuscript.
Round 2
Reviewer 3 Report
Comments and Suggestions for Authors
The authors, in their revised version, answered satisfactorily to reviewers points and suggestions, the only thing is lucking is about possible therapeutic perspective.